# Impact of Data and Study Characteristics on Microbiome Volatility Estimates

**DOI:** 10.3390/genes14010218

**Published:** 2023-01-14

**Authors:** Daniel J. Park, Anna M. Plantinga

**Affiliations:** 1Roivant Sciences, New York, NY 10036, USA; 2Department of Mathematics and Statistics, Williams College, Williamstown, MA 01267, USA

**Keywords:** microbiome volatility, longitudinal microbiome, temporal variability, qualitative changes, quantitative changes

## Abstract

The human microbiome is a dynamic community of bacteria, viruses, fungi, and other microorganisms. Both the composition of the microbiome (the microbes that are present and their relative abundances) and the temporal variability of the microbiome (the magnitude of changes in their composition across time, called volatility) has been associated with human health. However, the effect of unbalanced sampling intervals and differential read depth on the estimates of microbiome volatility has not been thoroughly assessed. Using four publicly available gut and vaginal microbiome time series, we subsampled the datasets to several sampling intervals and read depths and then compared additive, multiplicative, centered log ratio (CLR)-based, qualitative, and distance-based measures of microbiome volatility between the conditions. We find that longer sampling intervals are associated with larger quantitative measures of change (particularly for common taxa), but not with qualitative measures of change or distance-based volatility quantification. A lower sequencing read depth is associated with smaller multiplicative, CLR-based, and qualitative measures of change (particularly for less common taxa). Strategic subsampling may serve as a useful sensitivity analysis in unbalanced longitudinal studies investigating clinical associations with microbiome volatility.

## 1. Introduction

The human microbiome is a dynamic community of bacteria, viruses, fungi, and other microorganisms that has been associated with a wide range of human diseases, including irritable bowel disease, graft-versus-host disease, bacterial vaginosis, and Alzheimer’s [1,2,3,4]. With improvements in sequencing technologies and reductions in cost, longitudinal microbiome studies have become increasingly common in recent years. Longitudinal studies of microbiome composition have been used to investigate temporal changes in healthy individuals [5,6,7,8] and to understand the association of changes in microbiome composition with clinical changes during disease progression or treatment [9,10,11,12,13]. In addition to clinical associations with the abundance of specific taxa, microbiome volatility, which we define as the degree of compositional change over time [14], may have a separate and potentially bidirectional association with health [10,15].

Microbiome volatility does not have a single, formal mathematical definition. There are two major approaches to quantifying and comparing microbiome changes over time: distance-based and taxon-level approaches. In distance-based approaches, rather than defining an explicit volatility metric, investigators compute ecological distance metrics such as Bray–Curtis or UniFrac distances between samples [16], then compare the distributions of intraindividual and interindividual distances [14,15,17,18,19]. For example, Bastiaansen et al. [14] defined volatility as the Aitchison distance travelled over the course of a study and concluded that stressed mice had higher volatility than the controls; similarly, Halfvarson et al. [15] computed weighted and unweighted UniFrac distances between consecutive samples, set a boundary they called the “healthy plane” to represent the variability within healthy individuals, and found that participants with IBD had “considerable volatility away from the healthy plane”. Distance metrics may also be computed directly based on qualitative or quantitative taxon-level measures of change [20]. Formal taxon-level approaches to characterizing microbiome volatility are less common and typically require a long, dense microbiome time series. For example, measures related to microbiome volatility can be extracted from temporal models such as dynamic linear models [21] and the linear mixed model based approach called MTV-LMM [22]. The intraclass correlation coefficient has also been used to quantify stability [7,23]. In some cases, investigators combine these two conceptual approaches by associating the abundance of specific taxa with intraindividual variability [17].

However, there are several features of longitudinal microbiome studies that may pose challenges to volatility quantification and comparison. First, similar to most longitudinal studies, even microbiome studies that are designed to collect samples at balanced time points are afflicted by missing, mistimed, or QC-failed samples, so the sampling interval is rarely consistent for all intraindividual pairs of samples. In addition, sampling intervals for gut microbiome profiling are often tied to a biological mechanism such as a bowel movement, so timing cannot be precisely controlled by investigators. Second, the perennial difficulty posed by differences in sequencing read depth comes to the fore for comparisons of taxon presence across time points. The absence of rare taxa in a sequencing-based microbiome profile may represent true biological absence or insufficient sampling depth to detect the taxon. Hence, variability in the total read count across repeated samples may result in higher qualitative volatility estimates than would be supported based on a perfect knowledge of the microbial community. Third, microbiome data are compositional, so changes in a taxon’s relative abundance may be driven by changes in that taxon’s absolute abundance, but they may also be induced indirectly by changes in the abundance of other taxa.

Using four publicly available microbiome time series datasets, we investigated the potential impact of sampling interval length and sequencing read depth on several measures of change in microbiome composition. We used temporal subsampling and several rarefaction approaches to systematically vary the sampling interval and read depth, then computed additive, multiplicative, centered log ratio (CLR)-based, and qualitative (presence/absence) changes in the taxon abundance between time points along with the typical intraindividual distance-based analysis. This exploration reveals patterns in the measures of microbiome volatility across the study and sample characteristics, with implications for the design and analysis of microbiome volatility studies.

## 2. Materials and Methods

### 2.1. Datasets

The four microbiome time series datasets selected for use in this study varied in their sampling interval, overall study time frame, and body site. We focused on internal body sites (as opposed to the skin, for example, which directly interacts with the external environment) with different microbiome characteristics: the healthy gut microbiome tends to be diverse, whereas the healthy vaginal microbiome tends to have one or a few heavily dominant species. Selected study characteristics are summarized in Table 1 and visualized in Appendix A.

Caporaso et al. (2011) generated the densest and longest-running time series of the human gut microbiome to date, called the Moving Pictures study [5]. They sampled two healthy individuals almost daily for 6 months (female subject F4) or 15 months (male subject M3), sequencing the V4 region of the 16S rRNA gene to generate microbiome profiles. We used genus-level taxonomic profiles.

In the Student Microbiome Project (SMP), Flores et al. (2014) sampled 85 college-aged adults at four body sites (gut, forehead, palm, and tongue) weekly for 3 months, sequencing the V4 region of the 16S rRNA gene and clustering sequences into OTUs at 97% similarity [24]. We used only gut microbiome samples. Demographic, lifestyle, medication, and health status data were also collected weekly; subjects who took antibiotics during the study period were excluded from this analysis. Neither subjects nor samples were excluded on the basis of sickness during the study period (37 out of 58 subjects) or menstruation status. There were 10509 unique OTUs across all the samples, which we aggregated at the genus level for computational tractability.

Gajer et al. (2012) sampled the vaginal microbiome of 32 healthy women twice weekly for 16 weeks, sequencing the V1-V2 region of the 16S rRNA gene [25]. Participants reported menstrual bleeding, sexual activity, medications, contraceptives, and other characteristics in daily diaries. Taxon counts were computed based on the original study’s reported taxon proportions and total read counts, rounded to the nearest whole number. Taxonomic assignments were at the species level for the *Lactobacillus* species and genus level otherwise.

Ravel et al. (2013) characterized the vaginal microbiome daily for 10 weeks in 4 women without bacterial vaginosis (BV), 6 women with asymptomatic BV (ABV), and 15 women with symptomatic BV (SBV), sequencing the V1-V3 regions of the 16S rRNA gene and generating species-level taxonomic assignments [26]. For women with episodes of ABV or SBV, we included only samples prior to the BV episode. Notably, women later diagnosed with ABV or SBV typically had *Lactobacillus*-depleted vaginal microbiomes at earlier time points, despite the lack of an active BV diagnosis based on their Nugent score. We excluded women with fewer than 20 time points prior to their first active BV diagnosis.

### 2.2. Measures of Volatility

We considered five approaches to quantifying changes in the microbiome between pairs of time points. The four taxon-level measures of change were additive, multiplicative, CLR-based, and qualitative measures, as summarized in Table 2. We also considered intraindividual global dissimilarities.

Since the logarithm of zero is undefined, the CLR transformation requires replacing zeros with small non-zero values. There are a variety of methods for zero replacement [27,28,29,30]; we followed the common approach of adding a pseudocount of 1 to all counts [31]. Notably, $d^{c}$ is related to $d^{m}$ through a ratio of geometric means: (1)$$\begin{array}{ccc} d_{ijt_{k-1}t_{k}}^{c} & = & \log\left(\frac{{\tilde{p}}_{ijt_{k}}}{\mathrm{GM}({\tilde{p}}_{it_{k}})}\right)-\log\left(\frac{{\tilde{p}}_{ijt_{k-1}}}{\mathrm{GM}({\tilde{p}}_{it_{k-1}})}\right)\\ & = & \log\left(\frac{{\tilde{p}}_{ijt_{k}}}{{\tilde{p}}_{ijt_{k-1}}}/\frac{\mathrm{GM}({\tilde{p}}_{it_{k}})}{\mathrm{GM}({\tilde{p}}_{it_{k-1}})}\right)\\ & = & \log\left(d_{ijt_{k-1}t_{k}}^{m}\times \frac{\mathrm{GM}({\tilde{p}}_{it_{k}})}{\mathrm{GM}({\tilde{p}}_{it_{k-1}})}\right), \end{array}$$
where ${\tilde{p}}_{ijt}$ is the relative abundance of the taxon *j* for subject *i* at time *t* computed after pseudocount addition and GM() is the geometric mean. For taxa absent at both time points in a rarefied dataset,
(2)$${\tilde{p}}_{ijt_{k-1}}={\tilde{p}}_{ijt_{k}}=\frac{1}{\mathrm{Rarefied}\;\mathrm{Read}\;\mathrm{Count}\;+\;\mathrm{Num}.\;\mathrm{Taxa}}.$$
In this case, $d^{c}$ simplifies the log of the ratio of geometric means, so there is often a nonzero CLR-based difference even when the taxon is not detected at either time point.

Intraindividual dissimilarities were calculated using existing ecological distance metrics. For the Moving Pictures dataset, which includes a phylogenetic tree, we considered unweighted UniFrac, generalized UniFrac with α=0.5, weighted UniFrac, and Bray–Curtis dissimilarities. The UniFrac dissimilarities account for phylogenetic relationships among taxa, whereas Bray-Curtis only considers taxon abundance [16]. Unweighted UniFrac only uses presence/absence data; weighted UniFrac only uses abundance data; and generalized UniFrac is an intermediate version [32]. For the other three studies, because phylogenetic trees were unavailable, we only calculated the Bray–Curtis dissimilarity.

### 2.3. Sampling Interval and Sampling Depth Investigations

We considered four sampling intervals: 1 day, 3 days, 7±1 days, and 28±4 days, with the acceptable time lag between the samples designed to keep the observed sampling intervals to within 15% longer or shorter than the desired sampling interval. Once the pairs of samples at the desired sampling interval were identified, all samples were rarefied to the minimum read count in that study, the four taxon-level measures of change ($d^{a}$, $d^{m}$, $d^{c}$, $d^{q}$) were calculated for each taxon in each pair of samples, and the global dissimilarity was calculated between the sample pairs. Because the minimum read count was very small for the Ravel study, we treated 500 reads as the “minimum read count” for rarefaction purposes (leaving 5 samples unrarefied). The summary statistics included the standard deviation of $d^{a}$, $d^{m}$, and $d^{c}$ and the proportion of time pairs for which $d^{q}\neq 0$ were used to compare the magnitude of taxon-level changes based on sampling interval and average taxon relative abundance.

We also considered four rarefaction-based approaches to explore the impact of differential sampling depth. Specifically, we: (1) kept all of the original reads (no rarefaction), (2) adopted the conventional approach of rarefying to the minimum read count across all samples, (3) rarefied to 80% of the minimum read count, and (4) rarefied to 60% of the minimum read count. Using these four versions of each dataset with weekly sampling (7±1 day sampling interval), we repeated the summaries described above. As before, 100% of the minimum read count was considered 500 reads for the Ravel study.

Finally, to assess whether samples that originally had higher read counts had different estimated volatility even after rarefaction, we fixed the sampling interval for each study to a seven-day interval, rarefied to the minimum total read count, and calculated all measures of volatility. We averaged the two original read counts for each time point pair and explored associations of the taxon-level measures of change with the pairwise average read count.

## 3. Results

### 3.1. Sampling Interval Investigations

In the absence of systematic microbiome transitions to a new stable state over the course of a study, each measure of quantitative change (additive, multiplicative, CLR-based) should be centered around zero. For the most part, this is what Appendix A shows, although the CLR-based changes for abundant taxa at long sampling intervals (time lags) had a positive mean. Therefore, because comparing the centre of the distribution of log fold changes is not highly informative, we instead compared distributions based on the spread, where larger standard deviations (SDs) indicate larger increases and decreases in relative abundance.

The SDs for additive changes in relative abundance are shown in the top row of Figure 1 and in Appendix A. In all four studies, the overall SD was larger for longer time lags, and more abundant taxa had much larger SDs than rarer taxa. Taxa with relative abundances greater than 0.001 followed the overall association of higher SDs for longer sampling intervals. However, for rare taxa, the opposite was true: longer sampling intervals had smaller SDs.

For multiplicative changes, shown in Figure 1 (middle row) and Appendix A, the results were similar to additive changes. For rare taxa, smaller SDs were observed at longer time lags, whereas for common taxa, larger SDs were observed at longer time lags. The multiplicative SDs were shrunk less strongly towards zero than the additive SDs due to rare taxa that are absent at most pairs of time points, which have additive changes of zero but undefined multiplicative changes.

The results for CLR-based changes (bottom row of Figure 1; Appendix A) revealed consistent increases in SDs both with increasing taxon abundance and with increasing time lag within each relative abundance category. The two factors together had a synergistic effect: the ratio between the SDs of CLR-based changes for larger versus smaller time lags was greater for more abundant taxa.

Sampling interval was relatively unassociated with the probability of qualitative changes in taxon abundance, as seen in Figure 2. Qualitative changes were most common at intermediate taxon abundances, which is consistent with biological expectations: extremely rare taxa will almost never appear, and extremely common taxa will be present at nearly all time points, so qualitative changes are unlikely at both extremes. In the Moving Pictures dataset, the probability of qualitative change was noticeably higher for a 28-day lag than for the three shorter lag times, and similarly, for the Ravel dataset, the probability of qualitative change monotonically increased with increasing sampling interval. Although the SMP and Gajer datasets descriptively followed the same pattern, the differences are smaller.

Finally, for intraindividual global dissimilarities, there was a mild increase in the median Bray–Curtis dissimilarity between time point pairs as time lag increased for each of the four studies (Figure 3). The differences were most noticeable for the Moving Pictures dataset, particularly on day 28 (which was also the scenario with the highest rate of qualitative change). Both observations may result in part from the higher diversity of the Moving Pictures dataset, which has far more unique taxa than the other three datasets. To confirm that similar results hold in other common dissimilarity metrics, Appendix A compares four measures of global dissimilarity by time lag in the Moving Pictures dataset. Similar patterns across time lags were seen for each dissimilarity: the distribution of the dissimilarity value for lags of 1 and 3 days were similar, the distribution at 7 days had slightly fewer high outliers, and the distribution at a 28-day lag had higher variability (indicated by a larger interquartile range and heavier right tail).

### 3.2. Read Depth Investigations

Another key question in longitudinal microbiome studies is how much of an effect differences in read depth has on measured qualitative and quantitative change in microbiome composition and whether that effect persists even after rarefying the data to a common total read count. As in the previous section, the centre of each distribution of changes (with different rarefactions, at a fixed 7-day time lag) is zero, so we again compare SDs.

The SDs of **additive** changes in taxon relative abundance were similar regardless of taxon abundance category, study, and rarefaction procedure (top row of Figure 4; Appendix A). For **CLR-based** changes (bottom row of Figure 4; Appendix A), the SD decreased monotonically with increasing strength of rarefaction in each study and across all taxon abundance categories. As expected given the high variability in read counts within each study, the difference between no rarefaction and standard (100%) rarefaction is larger than differences between rarefaction levels. **Multiplicative** changes (middle row of Figure 4; Appendix A) matched CLR-based changes almost exactly, with the exception of rare taxa in the vaginal datasets (which often disappeared from the dataset entirely after rarefaction).

Differences in the frequency of qualitative change at each relative abundance occurred only between the unrarefied and rarefied versions of each dataset; the level of rarefaction did not make a notable difference (Figure 5). In the Moving Pictures, SMP, and Gajer studies, the shape of all four curves was similar, but the unrarefied curve was shifted to the left relative to the rarefied curves so that the maximum probability of qualitative change occurred at a lower abundance. The Ravel study showed a similar pattern but with larger standard errors around the LOESS curve. Because subsampling investigations focus only on sampling zeros (biologically present taxa that have zero counts due to undersampling), not structural zeros (zero counts due to biological absence), these results confirmed that more frequent intermittent sampling zeros, as seen with reduced read counts, were associated with higher estimated qualitative volatility.

The distribution of dissimilarities was nearly identical regardless of rarefaction choices, so mild shifts in the estimates of taxon abundance did not have much impact on the distance-based measures of intraindividual change over time (Figure 6 and Appendix A). The only noticeable differences in dissimilarity distributions occurred when unweighted UniFrac is used. In that case, the intraindividual dissimilarity tended to be very slightly higher with lower read counts (increasing rarefaction). Because read depth and rarefaction most strongly affect the presence of rare taxa, not the abundance of common taxa, unweighted UniFrac is by nature the most sensitive to read depth effects due to its exclusive use of presence/absence data.

#### Residual Effects of Read Depth after Rarefaction

Whether rarefaction as an approach to correct for differences in sequencing read depth is appropriate, inefficient but valid, or invalid is currently in contention (see, e.g., [33,34,35,36]). Some recent investigations have shown that there may be residual effects of differences in read depth even after rarefaction, particularly in analyses strongly affected by the presence of rare taxa, such as alpha diversity analyses [37]. Others show that rarefaction is inefficient but yields proper expected values for the desired quantities [34]. Because rarefaction is still commonly used in practice, we investigated whether there were residual effects of samples’ original read depths on measures of taxon-level change.

For additive changes, the top row of Figure 7 shows that in both gut studies (Moving Pictures, SMP), the SDs for taxa with relative abundance in (0.001, 1] were largest for the quartile of sample-pairs with the highest original read counts; however, in the vaginal studies (Gajer, Ravel), the samples with lower original read counts had higher SDs. Multiplicative changes also tended to have larger SDs for samples with higher original read counts in Moving Pictures, SMP, and Gajer (but not Ravel), for all but the rarest taxon abundance categories (Figure 7, middle row).

For CLR-based changes between consecutive non-zero time points, patterns differed between all four studies (Figure 7, bottom row). The Moving Pictures SDs tended to increase with increasing read count quartile. The SMP SDs tended to be highest for the samples with the lowest and highest quartile of original read counts. Gajer showed no association between SD and the original read count at all, and for Ravel, the samples in the lowest quartile of the original read count had the highest SDs.

Finally, for qualitative changes (Figure 8), the overall takeaways were similar to CLR-based findings. In all studies except Ravel, the samples in the highest read count quartile had the highest qualitative volatility; these differences were more noticeable in the Moving Pictures study than in SMP or Gajer. By contrast, in Ravel, the samples in the two lower read count quartiles tended to have higher qualitative volatility.

Because these potential residual effects were not consistent across the four studies, it is possible that some of the differences occurred by chance based on a particular subsampling of the data. Further investigation in additional time series studies is needed to clarify the extent to which residual read depth effects are a concern in volatility studies and perhaps suggest a better approach than rarefaction to handle them.

## 4. Discussion

Through systematic subsampling of four microbiome time series, we have explored the effect of sampling interval on measures of microbiome volatility. Quantitative measures of change, including additive differences in relative abundance, log fold changes in relative abundance, and additive differences in CLR-transformed abundance, tend to be larger at longer sampling intervals, particularly for common taxa. However, the increase in volatility is not constant across relative abundances, so simply using a multiplier such as the inverse time interval is unlikely to fully account for sampling interval-related differences. Sampling interval is generally unassociated with the probability of qualitative change, although in the Moving Pictures study, the proportion of qualitative change is greater at the longest time lags. The probability of qualitative change is largest for intermediate-abundance taxa in every study. Finally, the differences between the distributions of intraindividual dissimilarities across sampling intervals are minor in every study except Moving Pictures, for which dissimilarities are greatest at the 28-day time lag. Comparisons across time intervals are similar for the four distance metrics considered.

We have also explored the effect of read depth on each measure of change in microbiome composition through the rarefaction of each microbiome time series to different read depths with fixed 7-day sampling intervals. In general, additive measures of change are similar regardless of the read depth, whereas multiplicative and CLR-based measures of change are smaller with lower read depths. The distribution of the proportion of qualitative changes shifts to higher relative abundances with lower read depths. There are minimal differences in the distributions of distance-based measures of volatility with rarefaction.

Associations between the measures of change calculated on rarefied data and the original read count of the two samples are inconsistent across the four studies, so more evidence would be needed to conclude that there is a residual effect of sequencing read depth after rarefaction and to clarify what that effect is. However, ongoing investigations into this question are worthwhile, as results may vary depending on the distribution of taxon abundances, the measures being calculated post-rarefaction and their dependence on the presence of rare taxa, and other study factors.

Across all investigations, the distribution of differences in CLR-transformed relative abundances is relatively similar to the distribution of log fold-changes in relative abundances. Given the relationship between these two measures (Equation (Equation 1)), this behaviour is expected if the geometric mean taxon abundance is not extremely variable between the samples. Since using the CLR transformation mitigates the effects of compositionality in taxon-level analyses [38] and the requisite zero-adjustment allows absent taxa to contribute to multiplicative measures of volatility while maintaining results consistent with the raw log ratio among pairs of non-zero taxon abundances, the CLR-based approach is an attractive measure of quantitative microbiome volatility.

Taken as a whole, our results suggest that the effects of sequencing read depth and sampling interval are consistent across studies and body sites after accounting for taxon relative abundances. Sequencing read depth, consistent with general intuition, matters most for rare taxa and qualitative changes, and rarefaction helps avoid systematic differences in volatility estimation based on read depth. This comes at the cost of efficiency due to the reduction in the available data; more sophisticated methods, including some modern Bayesian approaches, estimate whether particular zeros are sampling or structural zeros and may avoid rarefaction (e.g., [30]). The sampling interval has potentially important effects on the average magnitude of quantitative and qualitative changes in taxon abundances. In the context of unbalanced studies (by design or through missing data), systematic differences in sampling intervals between clinical groups could impact the quantification of microbiome volatility and should be considered as a potential confounder. Despite resulting in a reduction in the total sample size, subsampling pairs of time points to a consistent time interval may serve as a valuable sensitivity analysis.

## Figures and Tables

**Figure 1 genes-14-00218-f001:**
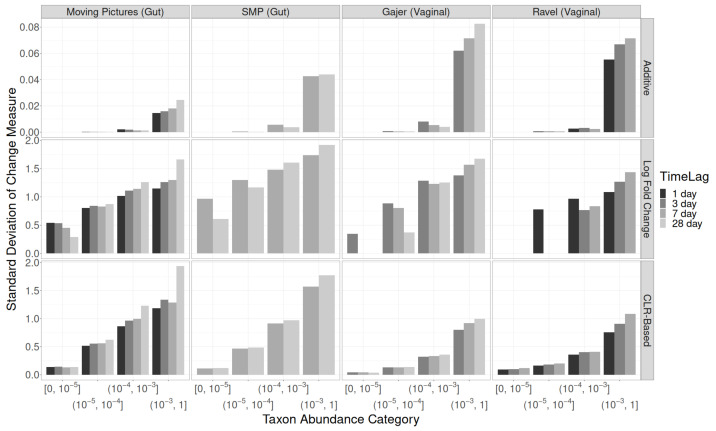
Standard deviations of quantitative taxon-level measures of change by time lag, taxon abundance category, and study. Top row: additive changes. Middle row: multiplicative changes (log fold changes). Bottom row: centered log ratio based changes.

**Figure 2 genes-14-00218-f002:**
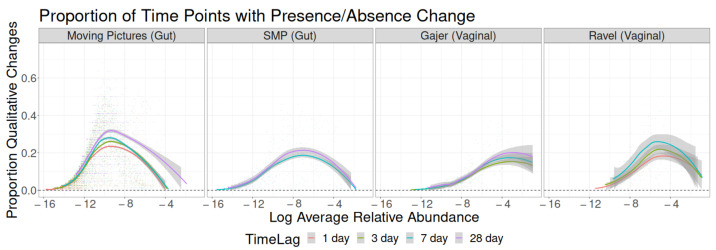
Proportion of time point pairs for which a taxon’s binary presence in the sample changes as a function of study, time lag, and the log of the taxon’s average relative abundance across all samples.

**Figure 3 genes-14-00218-f003:**
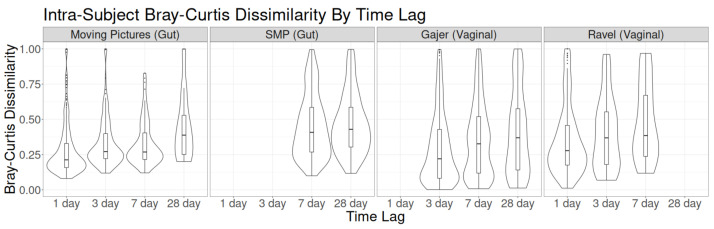
Distribution of intraindividual Bray–Curtis dissimilarity across time lags for each of the four studies.

**Figure 4 genes-14-00218-f004:**
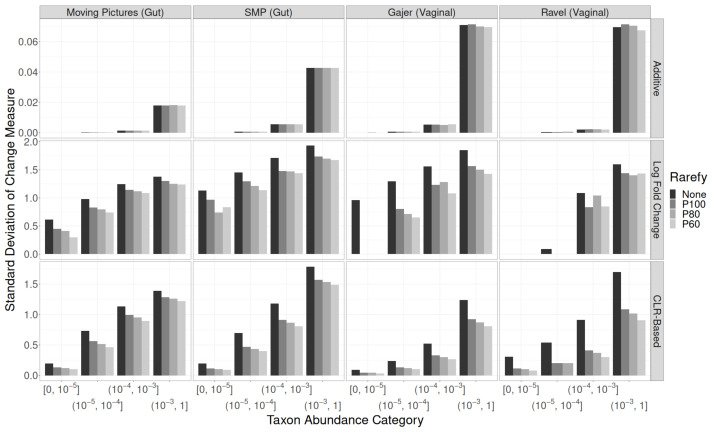
Standard deviations of quantitative taxon-level measures of change by rarefaction procedure, taxon abundance category, and study. Top row: additive changes. Middle row: multiplicative changes (log fold changes). Bottom row: centered log ratio based changes.

**Figure 5 genes-14-00218-f005:**
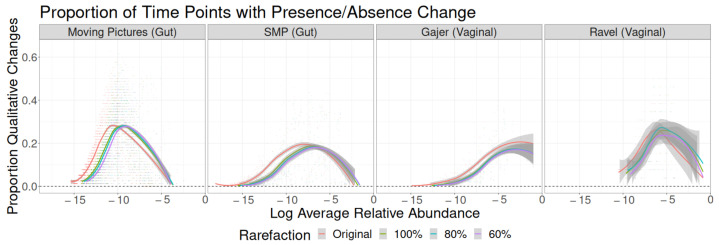
Proportion of time point pairs for which a taxon’s binary presence in the sample changes as a function of study, rarefaction approach, and the log of the taxon’s average relative abundance across all samples.

**Figure 6 genes-14-00218-f006:**
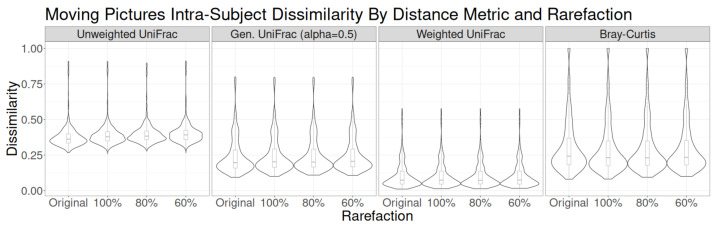
Intraindividual dissimilarity quantified using four metrics (unweighted UniFrac, generalized UniFrac, weighted UniFrac, and Bray–Curtis dissimilarity) for each rarefaction approach in the Moving Pictures study.

**Figure 7 genes-14-00218-f007:**
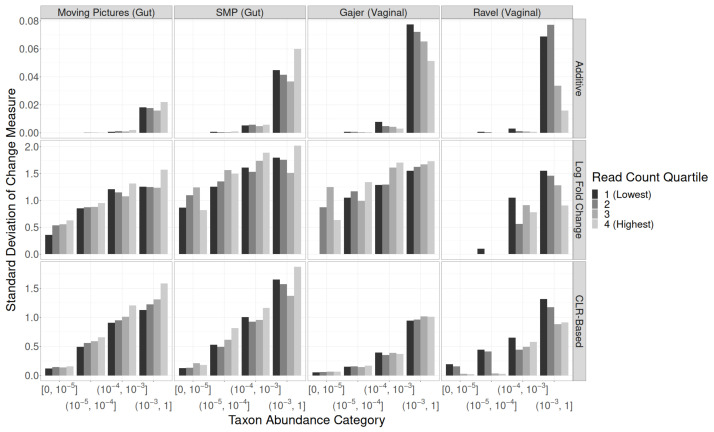
Standard deviations of quantitative taxon-level measures of change by original read count quartile, taxon abundance category, and study. Top row: additive changes. Middle row: multiplicative changes (log fold changes). Bottom row: centered log ratio based changes.

**Figure 8 genes-14-00218-f008:**
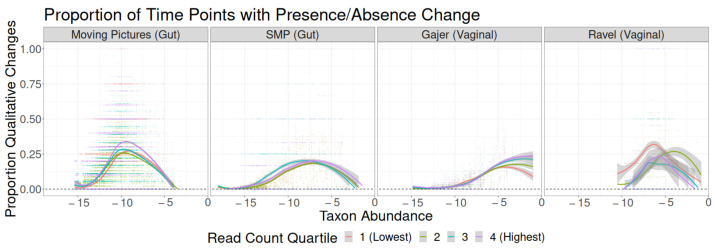
Proportion of time point pairs for which a taxon’s binary presence in the sample changes, as a function of study, original read count quartile, and taxon abundance quantile.

**Table 1 genes-14-00218-t001:** Characteristics of studies included in this investigation. Sample sizes and time points after all necessary exclusions.

	Caporaso et al., 2011 [5]	Flores et al., 2014 [24]	Gajer et al., 2012 [25]	Ravel et al., 2013 [26]
**Basic Study Information**				
Study name	Moving Pictures	SMP	-	-
Body site	Gut	Gut	Vagina	Vagina
Number of subjects	2	58	32	6
Percent female	50%	63.7%	100%	100%
Percent white	-	75.9%	40.6%	16.7%
Age (years): Mean (SD)	-	24.1 (6.4)	37.1 (8.1)	27.2 (6.3)
Age (years): Range	32–33	18–55	22–53	21–38
**Sampling Frequency and Study Duration**				
Number of time points	131–336	7–10	25–33	23–38
Sampling interval	Daily	Weekly	Twice-weekly	Daily
Study duration	6–15 months	3 months	16 weeks	10 weeks
**Summaries of Taxa and Reads**				
Read count: Median	36,114	43,282	2403	5195
Read count: Range	15,355–60,847	11,393–188,192	556–6619	145–15,972
Number of unique taxa	3962	632	331	122
Taxon analysis level	Genus	Genus	Genus/Species	Species

**Table 2 genes-14-00218-t002:** Taxon-level measures of change. For subject *i* and taxon *j* at consecutive time points, $t_{k-1}$ and $t_{k}$, $d_{ijt_{k-1}t_{k}}$ indicates the measure of change in taxon abundance between the two-time points. Relative abundance is indicated by $p_{ijt}$ and the centered log ratio (CLR) transformation is defined as $\mathrm{CLR}\left({\tilde{p}}_{ijt_{k}}\right)=\log\left({\tilde{p}}_{ijt_{k}}/\mathrm{GM}\left({\tilde{p}}_{it_{k}}\right)\right)$ where $\tilde{p}$ represent proportions computed after pseudo count addition, and GM() is the geometric mean.

	Definition	Requirements	Possible Values
Additive	$d_{ijt_{k-1}t_{k}}^{a}=p_{ijt_{k}}-p_{ijt_{k-1}}$	-	[−1,1]
Multiplicative	$d_{ijt_{k-1}t_{k}}^{m}=\log\left(p_{ijt_{k}}/p_{ijt_{k-1}}\right)$	$p_{ijt_{k}}>0,p_{ijt_{k-1}}>0$	(−∞,∞)
CLR-Based	$d_{ijt_{k-1}t_{k}}^{c}=\mathrm{CLR}\left({\tilde{p}}_{ijt_{k}}\right)-\mathrm{CLR}\left({\tilde{p}}_{ijt_{k-1}}\right)$	$\tilde{p}$ computed after zero-replacement	(−∞,∞)
Qualitative	$d_{ijt_{k-1}t_{k}}^{q}=I\left(p_{ijt_{k}}>0\right)-I\left(p_{ijt_{k-1}}>0\right)$	-	−1 (present → absent), 0,
			1 (absent → present)

## Data Availability

All data used in this manuscript are publicly available. Data access, processing, and the analysis code is posted on GitHub at https://github.com/aplantin/describing-volatility (manuscript version of code is 9 January 2023). The data processing code, prepared data, subsampling functions, and volatility estimation functions are also available as part of the R package MBVolDescrip in the same GitHub repository.

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
