# Peer review of "Impact of Data and Study Characteristics on Microbiome Volatility Estimates"

_genes, 2023, doi:10.3390/genes14010218_

Round 1

Reviewer 1 Report

Dynamic is an inherent feature of the microbiome and plays important role in human diseases and health. Unbalanced sampling intervals and different read depths are two common barriers in longitudinal microbiome studies. The studies regarding how they affect microbial dynamic analyses are limited. This study subsample the datasets from four public datasets to several sampling intervals and read depths and investigate the microbiome volatility between conditions. This study uncovers the effects of unbalanced sampling intervals and different read depths on the microbiome volatility to a certain degree. These findings are interesting. But the measurements regarding microbiome volatility and statistical analyses are kind of straightforward. This paper would have a big improvement if they would be addressed.

1. For the additive, multiplicative, and distance-based measurements, this paper conducts calculations at the taxon level one by one, using relative abundance. While compositionality is a common concern in the microbiome data, especially in the longitudinal microbiome data, as normalization is conducted at each time point, respectively. Does this study deal with compositionality?

2. Why does the study use the absolute additive change while the original multiplicative change is in the results? To me, the direct additive change makes more sense, compared to its absolute value. The multiplicative change shows that its mean keeps around 0. If directly using the additive change, what are the corresponding results?

3. In the multiplicative change, the authors excluded 0s from the calculation. I think the microbiome abundance that changes from a value to 0, or from 0 to a value indicates the dynamic. Excluding them would reduce more information, especially since microbiome data is very sparse. Can you add the corresponding analyses by adding a pseudo count to avoid 0s as a complement or a sensitivity analysis?

Author Response

We thank the Reviewer for these constructive comments. Our responses are in blue text below, and corresponding changes in the manuscript are highlighted in blue text as well.

Reviewer 1.

Dynamic is an inherent feature of the microbiome and plays an important role in human diseases and health. Unbalanced sampling intervals and different read depths are two common barriers in longitudinal microbiome studies. The studies regarding how they affect microbial dynamic analyses are limited. This study subsamples the datasets from four public datasets to several sampling intervals and read depths and investigates the microbiome volatility between conditions. This study uncovers the effects of unbalanced sampling intervals and different read depths on the microbiome volatility to a certain degree. These findings are interesting. But the measurements regarding microbiome volatility and statistical analyses are kind of straightforward. This paper would have a big improvement if they would be addressed.

  1. For the additive, multiplicative, and distance-based measurements, this paper conducts calculations at the taxon level one by one, using relative abundance, while compositionality is a common concern in the microbiome data, especially in the longitudinal microbiome data, as normalization is conducted at each time point, respectively. Does this study deal with compositionality?

This is an important observation. We have added comments about compositionality to the Introduction, Methods, and Discussion sections. Related to Question 3 below, we have also added results based on centered log ratio transformed abundances (after adding a pseudocount), which both avoids excluding zero-to-zero transitions in multiplicative measures of change and (at least in theory) addresses the problem of compositionality.

  1. Why does the study use the absolute additive change while the original multiplicative change is in the results? To me, the direct additive change makes more sense, compared to its absolute value. The multiplicative change shows that its mean keeps around 0. If directly using the additive change, what are the corresponding results?

As requested, we have changed the additive results in the paper to use the direct additive change rather than the absolute additive change. Correspondingly, we report standard deviations of additive changes (comparable to the results for log fold changes) rather than means, since the means are mostly centered around 0 (see Supplemental Figure 2).

One difference between the two approaches is that the standard deviation of direct additive changes is closely related to the average squared additive change, so it will place higher emphasis on large changes, whereas the mean absolute additive change (the original metric) will place less emphasis on large changes. Which component of the distribution should receive most weight is a scientific question with, as far as we can tell, no “right” answer.

  1. In the multiplicative change, the authors excluded 0s from the calculation. I think the microbiome abundance that changes from a value to 0, or from 0 to a value indicates the dynamic. Excluding them would reduce more information, especially since microbiome data is very sparse. Can you add the corresponding analyses by adding a pseudo count to avoid 0s as a complement or a sensitivity analysis?

We had originally intended the nonzero multiplicative changes to be used in combination with the qualitative change metric (proportion of time point pairs for which a change in taxon presence/absence was observed) for a more holistic view of volatility, related in spirit to two-stage models. However, the reviewer makes a good point that in practice, it is very common to use multiplicative analyses after adding a small pseudocount, and often by applying a CLR transformation to handle compositionality at the same time. We have now added a set of analyses and results taking this approach.

Reviewer 2 Report

In the manuscript, Park and Dr. Plantinga utilized four publicly available gut and vaginal microbiome time series, explored the pattern of microbiome volatility measures given multiple sampling strategies, and concluded that Strategic subsampling may serve as a useful sensitivity analysis in unbalanced longitudinal studies investigating clinical associations with microbiome volatility. The manuscript is too long with many details. The authors may consider removing redundant statements or moving details to supplementary materials. I have several comments are as follows:

1.     As authors mentioned that there is not a single formal mathematical Microbiome volatility definition. A brief review on reported measurements regarding the Microbiome volatility/stability is recommended.

2.     It would be suggested summarizing No. of participants, No. of time points, and other basic information in Table 1.

3.     A flowchart could be helpful to present the study population from 4 studies.

4.     Both OTUs and genera were mentioned in the manuscript. Was the analysis performed on OTU level?

5.     Why 4 types of distance-based changes were implemented on the Moving Pictures dataset, but for the other three studies, only Bray-Curtis dissimilarity was calculated?

6.     Please provide figures with higher resolution, especially for Figure2, where lines are overlapped.

7.     There are minor grammatical errors. For example, in line 131,” These changes are therefore are defined”. These Please try going through the manuscript and revise as much as possible.

Author Response

We thank the Reviewer for these constructive comments. Our responses are in blue text below, and corresponding changes in the manuscript are highlighted in blue text as well.

In the manuscript, Park and Dr. Plantinga utilized four publicly available gut and vaginal microbiome time series, explored the pattern of microbiome volatility measures given multiple sampling strategies, and concluded that strategic subsampling may serve as a useful sensitivity analysis in unbalanced longitudinal studies investigating clinical associations with microbiome volatility.

The manuscript is too long with many details. The authors may consider removing redundant statements or moving details to supplementary materials.

As requested, we have made the main text more concise and removed redundancy where possible. In particular, we described the measures of volatility more briefly by moving details into the new Table 2, moved the algorithmic details for sampling interval identification to the Supplement, converted the standard deviation results in the main text to bar charts, moved additional figures and tables to the Supplement, and revised for clarity and brevity throughout.

I have several comments are as follows:

  1. As authors mentioned that there is not a single formal mathematical microbiome volatility definition. A brief review on reported measurements regarding the microbiome volatility/stability is recommended.

We have restructured and expanded the Introduction to provide more detail about existing methods for quantifying and comparing volatility.

  1. It would be suggested summarizing No. of participants, No. of time points, and other basic information in Table 1.

We have added rows for the number of subjects and the range of number of time points to Table 1. Based on Question 3, we hypothesize that the Reviewer is also interested in other information about the study populations; to this end, we have added the percent of subjects who are female, percent of subjects who are white, age range, and mean (SD) of age. There are unfortunately not many demographic characteristics available from all four studies.

  1. A flowchart could be helpful to present the study population from 4 studies.

We are not sure what flow chart the Reviewer is envisioning; since we had no part in subject selection for the original studies, we cannot, for example, provide detailed flow charts about the selection process. We have added a graphical display of some of the information in Table 1 as Supplementary Figure 1, specifically body sites, time frame of study, sampling frequency, and sample size. We have also added more information about the study populations in Table 1, as mentioned above.

  1. Both OTUs and genera were mentioned in the manuscript. Was the analysis performed on OTU level?

The level of analysis differed by study, depending on what information was available. We have clarified the level of analysis in each study in Table 1 and improved the precision of our language.

  1. Why 4 types of distance-based changes were implemented on the Moving Pictures dataset, but for the other three studies, only Bray-Curtis dissimilarity was calculated?

We have access to a phylogenetic tree for the Moving Pictures study, but not for the other three studies. Therefore, we are unable to calculate phylogenetic distances such as the UniFrac distances for the other three studies. We have added a phrase to this effect in section 2.2.

  1. Please provide figures with higher resolution, especially for Figure 2, where lines are overlapped.

We have increased the resolution. Since the point of the original Figure 2 is simply that the lines are overlapping and centered around 0, we have moved this figure to the Supplement (Supplemental Figure 2) to streamline the manuscript. This also allows the figure to be viewed at a larger size.

  1. There are minor grammatical errors. For example, in line 131,” These changes are therefore are defined”. Please try going through the manuscript and revise as much as possible.

We have revised the manuscript for grammar and clarity of language.

Round 2

Reviewer 1 Report

The authors have addressed my concerns.